# Ischemic Stroke, Lessons from the Past towards Effective Preclinical Models

**DOI:** 10.3390/biomedicines10102561

**Published:** 2022-10-13

**Authors:** Beatriz Amado, Lúcia Melo, Raquel Pinto, Andrea Lobo, Pedro Barros, João R. Gomes

**Affiliations:** 1Molecular Neurobiology Group, IBMC—Instituto de Biologia Molecular e Celular, 4200-135 Porto, Portugal; 2i3S—Instituto de Investigação e Inovação em Saúde, Universidade do Porto, Rua Alfredo Allen, 208, 4200-135 Porto, Portugal; 3Scientific Writer, 4200-135 Porto, Portugal; 4Neurology Department, Centro Hospitalar de Vila Nova de Gaia/Espinho, 4434-502 Vila Nova de Gaia, Portugal; 5Stroke Unit, Centro Hospitalar de Vila Nova de Gaia/Espinho, 4434-502 Vila Nova de Gaia, Portugal

**Keywords:** ischemic stroke, cytoprotection, in vitro models, in vivo models, spheroids, organoids

## Abstract

Ischemic stroke is a leading cause of death worldwide, mainly in western countries. So far, approved therapies rely on reperfusion of the affected brain area, by intravenous thrombolysis or mechanical thrombectomy. The last approach constitutes a breakthrough in the field, by extending the therapeutic window to 16–24 h after stroke onset and reducing stroke mortality. The combination of pharmacological brain-protective strategies with reperfusion is the future of stroke therapy, aiming to reduce brain cell death and decrease patients’ disabilities. Recently, a brain-protective drug—nerinetide—reduced brain infarct and stroke mortality, and improved patients’ functional outcomes in clinical trials. The success of new therapies relies on bringing preclinical studies and clinical practice close together, by including a functional outcome assessment similar to clinical reality. In this review, we focused on recent upgrades of in vitro and in vivo stroke models for more accurate and effective evaluation of therapeutic strategies: from spheroids to organoids, in vitro models that include all brain cell types and allow high throughput drug screening, to advancements in in vivo preclinical mouse stroke models to mimic the clinical reality in surgical procedures, postsurgical care, and functional assessment.

## 1. Stroke

Stroke was described by the World Health Organization (WHO) in 1970 as a syndrome with “rapidly developing clinical signs of focal or global disturbance of cerebral function, lasting more than 24 h or leading to death, with no apparent cause other than of vascular origin” [1]. In recent years, the American Heart Association (AHA) and the American Stroke Association (ASA) consider this definition outdated, because it is mainly focused on clinical symptoms [2] and does not reflect significant advances in the “nature, timing, clinical recognition of stroke and its mimics, and imaging findings” [2]. As a result, in 2013, the AHA and ASA revised the stroke definition to include silent infarctions (cerebral, spinal, and retinal) and silent hemorrhages [1,2,3], and highlighted that stroke is both a clinical and a radiological diagnosis [1]. Nowadays, ischemic stroke is defined as a focal neurological deficit triggered by a vascular cause and traced to the central nervous system (CNS) [4]. The cessation of blood flow to the brain leads to an absence of oxygen and nutrients, causing extensive cell damage and neuronal death [5].

Stroke is the world’s second most common cause of death, and one of the major causes of adult disability and dementia [1,4,6,7]. Stroke events affect 13.7 million individuals per year worldwide, causing the death of around 5.5 million people each year, and constitute a huge global health burden [1,4,6,8]. Ischemic strokes account for about 71% of all strokes [9], and are caused by a reduction in blood flow caused by the occlusion of a brain artery [4,6,8]. The remaining 29% of stroke events are hemorrhagic originated by a cerebral artery rupture, and causing internal bleeding [4,6].

During the 1990–2016 period, the prevalence of stroke in low and middle-income countries doubled, while it decreased by 42% in high-income countries [10]. In 2017, the predicted cost for stroke management was €45 billion—including direct and indirect costs [11]. As reported by the Global Cost of Disease Study (GDB), the stroke survival rates improved, but the socio-economic burden increased, since treatments available reduced stroke mortality and increased the number of disabled [10,12]. A study estimates that by 2035, the number of strokes, globally, will increase by 34%, as populations grow and people live longer, with the number of survivors increasing by 25% [12]. Thus, it is essential to establish better prevention and management plans for stroke, especially recovery systems.

Primordial prevention of stroke focuses on educating the population [13], through educational campaigns, public lectures, and information stalls, to raise awareness of risk factors, healthy lifestyles, and stroke symptoms [12]. Although prevention of stroke also carries costs, it contributes to decreasing stroke rates [12].

## 2. Ischemic Stroke

Ischemic strokes are caused by thrombotic or embolic events that lead to the reduction of blood and oxygen supply to the brain [14,15]. The cause of the stroke has an impact on both prognosis and outcomes [14,15]: thrombosis, the most common, happens when a clot is formed in a vessel of the brain or neck [16]; embolism occurs when a clot moves from another part of the body to the brain [16].

Ischemic strokes can be classified as acute (AIS) or transient ischemic attacks (TIAs). AIS requires rapid and effective treatment to limit irreversible damage and persistent symptoms, TIA usually resolves in less than 24 h, by spontaneous reperfusion, and does not lead to significant damage or symptoms [16].

### 2.1. Risk Factors

More than 90% of the stroke burden is caused by modifiable risk factors [16]. An event of stroke can occur to any person, of any sex, age, ethnicity, or social class [17], but some individuals are more prone to stroke considering risk factors identified by epidemiological studies [18].

Non-modifiable risk factors include age, sex, race, and genetic factors [19]. Studies report that after 55 years of age, the risk of suffering a stroke event nearly doubles for each subsequent decade [18]. Regarding sex, the incidence in men is 33% higher than in women, although the severity is greater in women [20]. Concerning race/ethnicity, a study reports that in the USA, black individuals (both men and women) between 45 and 84 years old are three times more likely to suffer a stroke event than white individuals; this difference scatters beyond 85 years old [21]. Ischemic strokes can also be caused by highly penetrant Mendelian mutations [22,23], meaning that familiar history of stroke raises the probability that an individual will suffer a stroke [22,23]. Some of these loci are well-known [24,25], so patients carrying those mutations can be aware of the risk. Genetically caused strokes are typically early in onset and lack the usual risk factors [24,25]. Many of these non-modifiable risk factors harm the cardiovascular system, which is frequently the strongest indicator to assess stroke risk [26]. Hence, as people get older, atherosclerosis worsens, increasing their risk of suffering an ischemic stroke and myocardial infarction [27].

Modifiable risk factors include smoking, obesity, drinking alcohol, high-stress levels or anxiety, and a sedentary lifestyle [19]. Underlying disorders, such as hypertension, Diabetes Mellitus, heart/blood vessel diseases, or brain aneurysms [19] also take a role as stroke risk factors and should be monitored.

The single most important risk factor is hypertension [4]. In the United States, 46% of adults have high blood pressure [28], and in Europe, this risk factor is under-treated, with studies revealing disease control rates of 32% or less in people with established hypertension [12]. The WHO states that for every 10 stroke-caused deaths, four could be prevented if blood pressure was regulated [29]. A poor diet is the second leading factor related to stroke mortality, with diabetes and obesity accounting for the third and fourth most important risk factors, respectively [29]. Smoking is the fifth biggest risk factor—smokers display 4.5 times increased stroke mortality [29].

In recent years, significant advances in prevention plans and management of vascular risk factors, such as smoking cessation, and control of hypertension, hypercholesterolemia, and diabetes mellitus contributed to a decrease in stroke incidence [30]. Physical exercise reduces cardiovascular and cerebrovascular risk and boosts the expression of neuroprotective factors, effective to prevent stroke [26]. Concluding, it is critical to promote and support healthy behaviors, such as weight loss, a balanced diet, and frequent exercise [26,30].

### 2.2. Diagnosis

The most common signs of ischemic stroke are sudden numbness/weakness on one side of the body (hemiparesis) [4], confusion and difficulty speaking, trouble seeing with one or both eyes, sudden dizziness/loss of coordination, and acute headache with unknown cause [16,31]. Less common symptoms include nausea/vomiting, vertigo, and loss of conscience [4]. Stroke symptoms can be sudden (the most common), or develop slowly and progressively [32]. Time management of the signs and symptoms is extremely important, referred to as the time of onset (the time when signs first occurred, reported by the patient or observers) [33].

The patient’s prognosis may be improved if the ischemic stroke is diagnosed early [34]. Fortunately, the awareness of stroke’s signs and symptoms was popularized by some common methods. The face arm speech test (FAST), a rapid procedure intended to assist ambulance staff to recognize acute stroke, promotes early stroke detection and diagnosis [34]. A focused neurologic examination, ideally incorporating the National Institute of Health Stroke Scale (NIHSS), should be performed to determine stroke severity, recording the heart rate and rhythm, blood pressure, and the presence or absence of fever [30].

Following the first approach, imaging procedures should confirm stroke diagnosis, and distinguish the subtype of stroke, aiding with possible treatments [30,34]. In most institutes, non-contrast head computed tomography (NCT) is the first course of study [30,34]. This is a less expensive, quicker, and commonly available technique, compared to magnetic resonance imaging (MRI), but the extent and severity of the perfusion deficit are not provided by NCT [34]. Early MRI is useful to confirm stroke diagnosis, the time of onset, and the infarcted core [30]. Both ER and imaging services must be prepared for rapid screening, which is critical to define the appropriate treatment to save as much brain tissue as possible, allowing a better prognosis.

AIS management has changed drastically in recent years, with more individuals obtaining treatment to reduce death and long-term disability [35,36]. A key breakthrough was to create coordinated regional stroke care systems to promptly identify stroke patients and direct them to reference centers offering appropriate therapy [35,37,38]. This entails performing clinical and imaging evaluations by clinicians to identify the patients that are eligible for treatment [35,37,38]. Thus, stroke networks and protocols in healthcare systems are critical to ensure that patients receive prompt treatment.

### 2.3. Key Concept: Core and Penumbra

When an ischemic stroke occurs, blood flow and oxygen supply to the brain diminish rapidly [19] to a critical level [4]. In AIS, the parenchymal lesions can be divided into two morphological zones: the ischemic core and the penumbra [39,40]. The almost complete lack of perfusion defines the core area, characterized by dead or dying tissue in the center of the infarct—this site is irreversibly damaged [16,19]. In this region, the reduction of blood flow below 15–20% of regular levels causes neuronal necrotic death [16] that cannot be stopped by neuroprotective drugs [39]. Core size depends on occlusion rate, collateral blood flow, and how quickly reperfusion is achieved [4,16,41]. The ischemic core growth is closely related to a worse prognosis for patients [4].

The penumbra includes brain tissue that is partially perfused and degenerates more slowly [39,40]. This area is a dynamic zone, potentially salvageable, that displays loss of neuronal function but maintains cellular integrity [9,16]. The penumbra zone is subdivided into an apoptotic area (potentially salvageable) and live cells [42]. However, if not promptly reperfused, the penumbra will be incorporated into the ischemic core (Figure 1) [39,40]. To guide stroke therapy, it is necessary to estimate accurately the penumbra size, through modern imaging techniques, and to treat patients as soon as possible to reduce significantly penumbra extent, and prevent core growth [39,40].

### 2.4. Ischemic Stroke Pathophysiology

Although occlusion time determines the size of the injury and stroke outcomes, several pathways are triggered, even if reperfusion is achieved quickly [42]. Stroke’s physiological mechanisms are very complex and intertwined, with excitotoxicity, oxidative stress, and inflammatory processes playing a major role in neuronal damage [43].

The brain receives 20% of cardiac output at rest and is extremely vulnerable to ischemia. Even brief ischemic episodes can set off a complicated chain of events that can lead to lifelong cerebral damage [44]. Critically lowered cerebral blood flow (CBF) during brain ischemia, results in insufficient oxygen and glucose delivery, beginning the stroke pathogenic process (Figure 2) [44,45]. The lack of oxygen and nutrients impairs cellular metabolism and leads to a reduction in energy production, with consequent failure of energy-dependent systems, such as the sodium-potassium ATPase [44,46]. This ionic pump is crucial to maintain the ionic gradients across the neuronal membrane, meaning that when not working properly, the ionic imbalance leads to an increase in the release/inhibits the reuptake of excitatory neurotransmitters, such as glutamate [46]. Prolonged exposure to glutamate overstimulates ionotropic N-methyl-D-aspartic acid (NMDA) and 1-amino-3-hydroxy-5-methyl-4-isoxazole propionic acid (AMPA) receptors, causing an increase in calcium influx and leading to downstream activation of enzymes that breakdown cellular membranes, proteins, nucleic acids, and contribute to a buildup in oxidative stress [44,46]. These processes can culminate in neuronal death.

The mitochondria play a central role in metabolism, being responsible for energy production (ATP) via the electron transport chain, a process that also yields the release of reactive oxygen species (ROS). After ischemia, high levels of intracellular Ca^2+^, Na^+^, and adenosine diphosphate (ADP) induce excessive mitochondrial oxygen radical production [46], overwhelming endogenous scavenging mechanisms and causing direct damage to lipids, proteins, nucleic acids, and carbohydrates [44,45,46]. In particular, oxygen radicals and oxidative stress promote the formation of mitochondrial transition pores, compromising mitochondrial integrity, and leading to cytochrome c release (a trigger of apoptosis) [44,46]. Ischemia also increases the expression of pro-apoptotic Bc-l2 family members and p53 genes [24], activating caspases and calpain cascades that cause cellular apoptosis [24].

Post-ischemic inflammation is also a relevant process in stroke pathophysiology, with contributions from endothelial cells, astrocytes, microglia, and neurons [44,45,46]. Increased Ca^2+^, ROS production, and the lack of oxygen and nutrients can activate astrocytes and microglia. These cells produce pro-inflammatory cytokines, such as interleukin-1 (IL-1), tumor necrosis factor-1 (TNF-α), and interleukin-1β (IL-1β) [44,45,46], that reduce cell adhesion molecules and impair the function of the extracellular matrix. These events are critical to the infiltration of inflammatory cells into the brain parenchyma and for increasing brain inflammation [44,45].

Apoptosis is triggered by the generation of toxic mediators, produced by the activated inflammatory cells and damaged neurons [24].

However, the ischemic cascade also activates neuroprotective mechanisms that might limit apoptotic and necrotic cell death or display anti-inflammatory properties [24], such as increased expression of heat shock protein 70 (HSP70), Bcl-2 anti-apoptotic proteins, prion protein (PrP), neurotrophin-3 (NT-3), and the anti-inflammatory interleukin-10 (IL-10) [24].

### 2.5. Treatments

Currently, reperfusion of the ischemic brain remains the only available treatment. It consists in the restoration of the blood circulation and re-oxygenation of the affected tissues [47]. This treatment was introduced in 1995 [48], and despite occasionally having a contradictory effect (causing increased damage to the tissues) [47], it remains the most effective treatment for acute ischemic stroke [49,50]. The typical reperfusion technique provides a short time window for treatment, with only a small percentage of patients (usually 1% to 3%) receiving reperfusion therapy, due to a sizable risk of symptomatic hemorrhagic transformation [49].

Reperfusion therapies save penumbral tissue, reduce final infarct size, and enhance clinical outcomes, by restoring blood flow to vulnerable tissues before they advance into infarction [49,50]. The first approved reperfusion treatments consisted of thrombolytic methods—intravenous (IV) tissue plasminogen activator (tPA) [50]. Only in 2015, after years of research and clinical trials [51], endovascular techniques were approved as a reperfusion method—relying on methods such as mechanical thrombectomy (MT) [50].

#### 2.5.1. Thrombolysis

Thrombolysis is a pharmacological treatment that involves the infusion of tissue plasminogen activator—tPA, or a tPA analog—to promote the lysis of the blood clot via the activation of the proteolytic enzyme plasminogen into plasmin [50]. Plasmin cleaves cross-link connections between fibrin molecules, which provide the structural scaffold for blood clots [50]. Clots became soluble and are further degraded by other enzymes, allowing blood flow to be restored [50]. The three principal types of plasminogen activators include tPA, streptokinase (SK), and urokinase (UK), and they differ in the mechanism of action on fibrin molecules [50]. Alteplase (recombinant tPA—rtPA), Retaplase, and Tenecteplase (TNK-tPA) are examples of recognized tPA analogues [50].

In 1995, a National Institute of Neurological Disorders and Stroke clinical trial changed AIS treatment, by demonstrating the safety and efficacy of intravenous (IV)-tPA [51,52]. It demonstrated that when IV-tPA is administered up to 3 h after symptom onset, patients are around 30% more likely to have only minor to no disability at 90 days following stroke [51,52]. The limited time window for IV-tPA usage in ischemic strokes makes it available to only 3.2–5.2% of AIS patients in the USA [51,52]. The AHA/ASA amplified the IV-tPA window from 3 to 4.5 h in 2009, based on European research, increasing the usage of this treatment to 20% of the patients [52].

Thus far, alteplase is the only thrombolytic drug approved by the Food and Drug Administration (FDA) for the treatment of AIS [50]. Recent advances were made with other thrombolytic drugs, such as tenecteplase or desmoplase, both in [41] Phase III trials (in 2020 and 2018, respectively). These therapies display several advantages over alteplase: [41] tenecteplase shows a reduced risk of hemorrhagic transformation [41]; desmoplase can be safely administered up to 9h after symptoms onset [41]. Furthermore, a recent trial (EXTEND-IA TNK) demonstrated that tenecteplase leads to better functional outcomes than alteplase [53]. Tenecteplase was introduced to stroke guidelines as an alternative to alteplase after these findings, and some stroke units use it frequently as a substitute for alteplase [54,55].

#### 2.5.2. Mechanical Thrombectomy

Mechanical thrombectomy (MT) is another extensively used treatment, after the publication in 2015 of five clinical trials showing benefits in MT up to 6 h after stroke symptoms onset [51,52]. Multiple trials have indicated that MT, in addition to normal medical therapy, improves the overall outcome of AIS patients with occlusion of the proximal middle cerebral artery (MCA) or internal carotid artery (ICA), when treated until 24 h after symptom onset [51,52]. Thanks to MT, the temporal frame for AIS treatment has been extended, providing clinicians with more robust therapeutic armament [52]. Modern MT more than doubles the probability of a better functional outcome compared to standard therapy alone, according to a pooled meta-analysis, with no significant difference in mortality or risk of parenchymal hemorrhage after 90 days [52].

Instead of breaking down the clot with chemicals, microcatheter devices (stents) are used in mechanical thrombectomy to remove the blood clot from an occluded artery with minimally invasive surgery [50,51]. This technique provides a larger time window than thrombolysis, with fewer adverse effects [51,56]. Additionally, when stents cannot access some brain areas or the clot is not easily removed, the aspiration technique might be used, [56] to aspirate the clot with a catheter and retrieve it [56].

Recent developments in the field of MT consist of combining stent retriever techniques with aspiration [57], increasing the rates of successful reperfusions. Furthermore, the establishment of guidelines for anesthesiology (as presented in the GASS trial [58]) and post-thrombectomy management [57] are important factors to increase the success of this technique. The Mechanical Thrombectomy with or without Intravenous Alteplase in Acute Stroke trial is the first study that discusses the hypothesis that mechanical thrombectomy alone might not be inferior to the combination of mechanical thrombectomy with intravenous alteplase, in patients eligible for both treatments. Previously, several observational cohort studies have investigated the outcome of intravenous thrombolysis before mechanical thrombectomy [59,60,61]. A study comprising 656 patients in China shows that mechanical thrombectomy alone within 4.5h of stroke onset is similar to mechanical thrombectomy preceded by intravenous alteplase, in a three-month functional follow-up [59,62].

Concluding, thrombolysis and mechanical thrombectomy are used to stop ischemia and restore the brain’s blood supply, but they do not limit the brain damage caused by inflammation and other processes during reperfusion [51]. Thus, pharmacological therapies that limit cell damage are required to improve stroke treatments.

#### 2.5.3. Neuroprotective Drugs

The primary purpose of therapeutic intervention in stroke is to reduce lethality and comorbidities associated, leading to good neurological outcomes for the survivors—in terms of motor coordination, memory, and speech [63]. However, even with successful reperfusion, not all patients recover their motor and neurological function [4]. Therefore, the development of novel neuroprotective therapies is a growing research field.

There is a massive effort from the scientific community to find pharmacological therapies that protect the cerebral tissue after brain ischemia [64]. For decades, the research focused on neuroprotection, considering that neurons were the main affected cells by ischemia [63,65]. Nowadays, it is considered that glial cells also take part in ischemic stroke pathophysiology [65], mainly in the inflammatory response. Hence, a more comprehensive approach to the development of brain protective drugs was established, including the study of oligodendrocytes, astrocytes, and microglial cells, as well as neurons [66].

The recent concept of the neurovascular unit (NVU), which integrates glial cells, neurons, vascular cells, and the basal matrix, was a major milestone in brain-protective research [67]. The focus is to develop therapies that mitigate the damage after stroke and diminish/ reverse apoptosis in the penumbra area, impacting all cell types comprised in the NVU [68]. Several studies indicate that reperfusion therapies combined with cytoprotective drugs can extend the therapeutic window, and reduce the damage to the blood–brain barrier (BBB) and the risk of hemorrhagic transformation post-ischemia [38,63]. Moreover, the recent discovery of diverse patterns for cell death and reperfusion is opening the doors for discoveries regarding cytoprotectants [69]. Cytoprotective drugs are now complementary to reperfusion, and not administered alone, which is a new step for successful therapies [70].

Due to the complexity of the events occurring during stroke, several types of protective drugs were investigated (Table 1). The first type centered on the blockage of voltage-gated cation channels [63], especially calcium channels [63,68]. Several drugs targeting this mechanism were tested. Nimodipine, one of the first candidates to go through clinical trials [71], blocks L-type calcium channels, preventing calcium influx and vasoconstriction [72]. The VENUS trial, a phase III clinical trial, tested nimodipine 6 hr after stroke onset [73], but was terminated prematurely since it was not effective [71]. A small Phase I clinical trial (MAVARIC) is investigating the efficacy and safety of verapamil (a calcium channel blocker) and magnesium administration after reperfusion [74]. The results are yet to be published, but another trial demonstrated that magnesium sulfate did not improve functional outcomes, when administrated alone [75].

Therapies targeting glutamate excitotoxicity, one of the main processes in stroke pathophysiology, were also tested. An example is N-methyl-D-aspartic acid receptor (NMDAR) antagonists. NMDARs are glutamate receptors, which are overactivated during excitotoxicity [63,68] and contribute to neuronal death [49]. The inhibition of these receptors using, for instance, MK801, reduces the neuronal response to glutamate and brain infarct; in animal models, MK801 reduced infarct volume by up to 75% [68]. However, this drug blocks the whole excitatory pathway, causing side effects such as memory loss [68], and has a short therapeutic window, which stopped its progression to clinical practice [76]. Nerinetide (aka. NA-1), a therapy targeting excitotoxic mechanisms was developed recently. The post-synaptic density protein 95 (PSD-95) is a scaffolding molecule [77] that binds NMDAR, triggering different intracellular pathways [78]. In excitotoxic conditions, such as the ones occurring in stroke, PSD95-NMDARs interaction causes Ca^2+^ overflow, leading to increased production of nitric oxide (NO), and activating pro-apoptotic factors [78]. Thus, NA-1 interferes with the PSD-95 [72] interaction with NDMARs and suppresses this excitotoxic pathway [79]. In macaque models of MCA occlusion, this drug was demonstrated to be benefic, reducing infarct volume, so it advanced to a Phase III clinical trial. ESCAPE-NA1 trial proved that nerinetide reduced mortality, improved clinical outcomes, and reduced infarct volume [80]—based on these outcomes, this therapy was approved by the FDA [81]. However, in patients receiving intravenous thrombolytic agents therapy, nerinetide was not effective [80], since tPA produces plasmin that cleaves and inactivates nerinetide [81]. A new trial, ESCAPE-NEXT, aims to combine nerinetide and mechanical thrombectomy for patients with acute ischemic stroke [82]. This trial is expected to be completed in August 2023, with outcomes assessed 90 days after the drug administration [82].

Another therapeutic target to prevent stroke-induced cell damage is the suppression of apoptosis [71], by inhibiting apoptotic proteins from several pathways. Despite conferring cell protection, the effects were transient and cell death occurred afterward [42]. Some researchers believe that molecules from a higher level of regulation should be considered: for example, upregulation of the angiotensin II type 2 receptor increases the expression of brain-derived neurotrophic factor, which can reduce caspase-3 activity, and consequently apoptosis [83]. Despite this, no apoptotic inhibitor has reached clinical trials, since the mechanism of action is irreversible, and has poor brain penetration [71].

Free radicals production is a major cause of stroke damage [68]. Free radicals are unpaired electron molecules, such as ROS, that cause cell death due to their instability and reactivity [63]. The use of free radical scavengers and antioxidants may protect the brain cells after a stroke [68]. NADPH-oxidase (NOX) inhibitors, as well as other free radical scavengers, interact with ROS, converting them into stable molecules [68,84]. Uric acid acts as an antioxidant agent and reduces brain swelling, inflammation, and apoptotic death [85]. Despite the uric acid mechanism of action is not fully understood [85], it evolved into Phase III clinical trial (URICO-ICTUS) [85], where it improved patients’ functional outcomes. Nonetheless, the sample size was restricted (421 patients) [86], and a confirmatory trial to further test uric acid as a potential stroke therapy is planned [87]. Additionally, novel antioxidant compounds are emerging, such as NBP, a synthetic compound, now in a Phase III clinical trial (BAST) predicted to end in December 2022 [88]. Previously tested free radical scavengers, as described by Ebselen [89] or Edaravone [90] should be reconsidered and tested under new clinical trial guidelines for AIS (combined with reperfusion, for example).

Anti-inflammatory agents may also be used to reduce the inflammatory process after stroke [68]. These agents are divided into categories: pro-inflammatory cytokine inhibitors; AMPK regulators/activators; and chemokines antagonists [68]. A case of success is the human urinary glycoprotein kallidinogenase (HUK), an anti-inflammatory agent tested in animal models [91]. HUK reduced stroke infarct volume and improved neurological outcomes [92]. This drug was approved by China’s regulatory authorities [51], and is being evaluated in a Phase IV Clinical Trial [93].

The recent discovery that a combination between cytoprotectants and reperfusion might be the way to achieve favorable outcomes brought back some drugs to the table: previously discarded compounds might now be re-evaluated under this new light [94]. A 2021 review article presented an overview of multiple drugs that underwent clinical trials previously and failed; the present review gives new insight into which drugs should be reconsidered in combination with reperfusion therapies [94]. Promising results for oxidative stress inhibitors, such as Edaravone, or promoters of neurogenesis, like NBP or NTP, indicate that these drugs should be re-purposed for clinical trials in combination with EVT [94]. This review also displays safety concerns regarding blockers of NDMA/AMPA receptors, such as Aptiganel HCl and Fanapanel [94]. Since patients present different lesions, compounds that target multiple mechanisms or a combination of different compounds should be addressed shortly [94].

Other molecules showing promising results include naturally occurring molecules, such as transthyretin (TTR). TTR has neuroprotective effects [95], promoting neurite outgrowth and increasing neuronal survival [96]. This ultimately could lead to better outcomes in motor function and memory [95].

New targets are recently being addressed, with special emphasis on cell-based therapies. The ability of stem cells to differentiate into different cell types contributes to the recovery of destroyed cells after AIS [97]. Additionally, these cells secrete neurotrophic factors and provide anti-inflammatory agents in the stroke’s aftermath, making them a great innovative therapy for AIS [98]. Previous phase I/II trials with allogeneic stem cells were completed, using adult mesenchymal bone marrow stem cells [99] to achieve this purpose. In this trial, patients received a single dose of allogenic stem cells, and behavioral endpoints were assessed 12 months later—behavioral improvements suggest this therapy should be further studied [99]. Recent phase I/II trials researching other methodologies for stem cells are emerging. One example is J-REPAIR, a phase I/II clinical trial with JTR-161, an allogeneic stem cell product from human dental pulp [100]. This clinical trial aims to assess the outcome for AIS patients (using the mRS, NIHSS and Barthel Index), by evaluating the safety and efficacy of this cell-based therapy.

Immune responses triggered by ischemia are usually involved in the worst outcomes, due to their role in the formation of edema and possible hemorrhagic transformation [101]. Thus, immune modulators are target for the treatment of AIS. A phase II clinical trial from 2015 tested Fingolimod, in conjunction with alteplase [102], and concluded that the administration of alteplase with the immune modulator reduced injury, and improved outcomes. More recently, in phase I/II trial, alteplase was administered with dimethyl fumarate, another immune modulator [103]. This trial is assessing the same primary outcomes and it is still recruiting, with the last updates from June 2022.

In the past, a considerable number of protective compounds were studied in vitro and in vivo with promising results, but they failed in the clinical trials [68]. There seems to be a roadblock between stroke models (especially animal models) and humans, due to differences in brain function/structure, and the individuality of each animal, among other factors [68]. Additionally, evidence shows that protection might depend on the type of cerebral ischemia and injury [104]. As previously stated, therapies usually focused on a single brain cell population (instead of the NVU), delaying the discovery of effective drugs [68]. In the upcoming years, new clinical trials should arise, to test drugs that confer protection to different cell types when combined with reperfusion therapies, despite their mechanism of action [69].

Despite the failures in translation from bench to bedside, there is a dim light in cell protectants research: more drugs are heading to clinical trials, proving to be safe and approved by some countries’ regulatory agencies (such as China’s). Additionally, with advances such as mechanical thrombectomy, clinical trials are considering the reperfusion state of each patient, as well as practicing more rigorous laboratory/preclinical guidelines [87].

### 2.6. Models for Acute Ischemic Stroke

A significant repertory of models is used to address specific issues of ischemic stroke, considering stroke’s varied etiology and wide range of manifestations [16,105]. The fact that, except for the recanalization of the occluded vessel, few experimental protective drugs were translated into the clinical practice so far exemplifies the gap between preclinical stroke models and the reality of human patients [16,105].

It is critical to understand the strengths and limitations of each stroke model, and select the model for each study, considering the aim of the study and the specific aspects of ischemic stroke that need to be addressed [16,106].

#### 2.6.1. In Vitro Models for AIS

In vitro stroke models are used to investigate specific molecular pathways occurring in stroke pathology. These models present many advantages: are reproducible, easy to produce and maintain, and cost-effective. These characteristics qualify them for high throughput screenings (HTS) to test therapeutic compounds’ efficacy [16,107].

In vitro models include organotypic brain slices and dissociated cultures of brain cells [16]. The slices keep the integrity and morphology of a specific brain region, including the visual map of the brain vessels, and mimic pathophysiologic processes, such as depolarization and penumbral area [16]. However, brain slices have to be perfused with artificial CSF, and can only be maintained for less than 12 h [107], limiting the time window for experimental procedures. Regarding dissociated brain cell cultures (Figure 3), they can be monolayers or co-cultures of various cell types, obtained from primary tissue [26]—directly isolated from the parental tissue—or immortalized cell lines [108]. While cell lines are easier to maintain, and have a prolonged life span, primary cells resemble more closely the parental tissue and the behavior of an in vivo environment [109], providing a more relevant tool for stroke studies.

Two major models simulate stroke in vitro: oxygen and glucose deprivation (OGD) and the chemical/enzymatic blockage of metabolism [16]. The OGD model, the most used, consists on the replacement of the equilibrated O_2_/CO_2_ medium with a N_2_/CO_2_ medium, in a hypoxia chamber [107]. Hypoxia alone can cause cellular dramatic changes, but usually, this model associates the lack of oxygen with the absence of glucose [110]. Cell cultures can be exposed to OGD for 1 to 24 h [16], with half-maximal neuronal loss occurring between 4 and 8 h of exposure to low oxygen and glucose [111]. Reperfusion can be simulated by returning to the initial conditions, with a glucose-containing medium in the presence of oxygen [16]. When cells undergo OGD, the effects are consistent with in vivo observations: swelling, apoptosis, necrotic death, and excitotoxicity [110,112,113].

Another model that mimics stroke in vitro is chemical or enzymatic blockage of the cellular metabolism [16], using compounds to inhibit specific pathways, such as antimycin to inhibit the electron transport chain, therefore, impairing cellular energy production [114]; or compounds to block specific enzymatic systems, such as 2-deoxyglucose, which blocks glycolysis [107]. These methods are cheap and easy to use, but the OGD model is more comparable to in vivo stroke effects, since chemical inhibition may cause exacerbated or collateral effects [107].

In recent years, monolayer 2D cultures evolved into 3D cultures containing different cell types, to better mimic the complexity of living organisms. These models allow cells to aggregate in a 3D shape, and cell–cell and cell–extracellular matrix interactions, mimicking brain tissue [26,107,115]. Furthermore, 3D cultures are compatible with high throughput screening and aid in the study of mechanisms involving each cell type during ischemia [116], and 3D cultures closely mimic physiological behavior, presenting a compact, organized structure, with gene expression, protein production, and dynamic equilibrium.

Early 3D cultures required a scaffold to develop into complex and organized subjects for study [117], such as hydrogels or porous polymers [118]. In recent models—organoids and spheroids—cells self-assemble and aggregate, creating their extracellular matrix, to resemble human tissues, and no external matrix is required [117,118].

However, it is important to remember that 3D cultures also present disadvantages: they are expensive, time-consuming, and have limited analyzing methods so far [119].

Spheroids and organoids present several differences, each one with its advantages and disadvantages, displayed in Table 2.

Spheroids were developed in 1970 and have been improved ever since [119]. These sphere-shaped cultures are formed using differentiated or undifferentiated cells that self-assemble, create an extracellular matrix, differentiate at the molecular level, and respond to external stimuli [120]. These models can be obtained using the hanging-drop technique or non-adhesive multi-well plates [120]—both methods are appropriate for high throughput screening [120], representing an advantage in the research of brain-protective drugs. Additionally, spheroids can be produced from different cell types, using human cells (or cells from other animals), to generate a multicellular organization similar to the tissue of interest [120]. Spheroid use has several drawbacks: a fragile structure requiring careful handled to maintain integrity; low homogeneity; different properties in the interior, such as an acidic environment, that may condition the action of a therapeutic drug; and lack appropriate analysis methods [119,121].

Organoids are also 3D matrix-free structures that differ from spheroids in the sense that they derive from organ-specific stem cells [120], either embryonic, pluripotent, or adult [119,121]. Stem cells aggregate in a spatial structure in such a way that they will differentiate and self-renew [120]. Organoids can be produced using a laminin-rich basement membrane extract combined with growth factors [119], or using spinning bioreactors [119]. This model allows the co-culture of several cell types, with cell–cell and cell–extracellular matrix interactions [121], and is more stable than spheroids, keeping its shape due to a more rigid matrix [121]. On the other hand, organoids are more expensive, have low reproducibility, and might condition the therapeutic responses to the formed matrix [121]. Hence, despite the advantages of this model for the study of stroke’s pathophysiology, it is not ideal to address the effect of protective drugs, since it cannot be used for high throughput analysis.

For studies involving CNS pathologies, particularly stroke, spheroids are the most used 3D in vitro model. To date, few studies evaluate the damage caused by OGD in spheroid cultures. In a recent study, OGD was used to simulate stroke in spheroids produced from the mouse brain cortex [122]. Researchers evaluated the expression of some key proteins that are associated with stroke effects (S100B, interleukin-1β, and myelin binding protein). Thus, it proved that the OGD model of in vitro ischemia successfully works on cortical spheroids [122], opening gates for new research on stroke in vitro.

In cancer studies, spheroid models were used to evaluate the effect of putative therapeutic compounds, by assessing spheroid morphology, using for instance image cytometers [123,124]. One paper used fluorescent markers—PI and caspase 3/7—to determine cell viability and apoptosis by fluorescent microscopy in spheroids. This contributed to establishing this method as a tool to investigate drug exposure time points, and kinetic/apoptotic drug effects [123], also in stroke studies. Another paper presented several microscopy, quantitative and molecular biology approaches to evaluate the effect of certain drugs on spheroids, providing useful information for the use of spheroid models in stroke research [124].

Additionally, spheroids can be used to study different features that might be important for HTS assays. Nzou, G. et al., for example, developed a brain cortical spheroid model to study tight junctions and the formation of the blood–brain barrier (BBB) [125]. BBB in vitro models are highly important to guarantee advances in drug screening [126], considering that not all drugs can penetrate the BBB. In another paper, Nzou G. et al. use the same model to assess the effects of anti-inflammatory agents/free radical scavengers in hypoxia, regarding drug delivery. These studies are important to validate 3D spheroids suited to assess potential therapeutic strategies for stroke and other brain pathologies.

The establishment of forwarding, easy-to-perform procedures that aid in in vitro stroke damage assessment (and therapeutic effects) is now the goal. The use of 3D models to better recapitulate the brain environment and allow for high throughput screenings improve the chances of finding new successful therapies. The main endpoint is to develop cytoprotective therapies that can be combined with reperfusion therapies or regenerating drugs. If these studies provide promising results, they should evolve into more complex models in vivo.

#### 2.6.2. In Vivo Models for Acute Ischemic Stroke

The first focal cerebral ischemia model in rodents was described in the early 1980s and has been widely used ever since [127,128]. Stroke studies using rodent models have several advantages: are predictable and reproducible; allow access to brain tissue for pathophysiological research, required to study the intricate biochemical and molecular pathways that comprise the “ischemic cascade” and contribute to cell death; deliver information on cerebral blood flow, to establish the potentially salvageable tissue-penumbra; test innovative therapies to reduce ischemic injury, save penumbra tissue, and promote long-term repair and recovery [129,130,131]. The majority of in vivo stroke models use mainly young adult male mice, but studies involving both sexes, older animals, and including other risk factors and comorbidities are increasingly common in the literature because they are more representative of human stroke patients [132].

When attempting to mimic the heterogeneous nature of stroke in humans, no animal model is a perfect representation of the disease, and each model has its strengths and weaknesses [133].

The middle cerebral artery (MCA) and its branches are the most often damaged cerebral vessels in human ischemia, accounting for over 70% of strokes [133]. Thus, the intraluminal suture MCA occlusion model is the most commonly used experimental model in rodents (Table 3A) [134]. Briefly, to perform this model, a filament is placed at the origin of the MCA and is maintained during the total occlusion period [133,135]. This model enables the precise control of occlusion time, for either transient ischemia, where the filament is removed after the desired occlusion time allowing reperfusion of the MCA area, or permanent ischemia, where the filament is left in place [133,135]. Vessel access does not require craniotomy, avoiding cranial trauma, which could affect intracranial pressure and post-stroke outcomes including reduced lesion volume [136,137]. The MCAO model has some differences when compared to human strokes: generates large infarct sizes surrounding the striatum and cortex, but can also result in hypothalamic injury, which is uncommon in humans [138]; represents an all-or-nothing strategy by inserting the intraluminal filament into the MCA’s origin, which is not illustrative of clinical reality, in which human strokes are frequently not caused by complete occlusions [133,139]; does not replicate the occurrence of partial spontaneous reperfusion, which can occur within 48 h after stroke onset [133,139]; does not depict the gradual clot breakdown following rtPA injection, but rather depicts spike reperfusion upon filament removal [140].

Recently, with the introduction of mechanical thrombectomy in the clinic, the MCAO model has gained more significance [141]. In fact, in 2015, five randomized controlled clinical trials that treated patients with major vessel occlusions either with or without rtPA treatment indicated positive results (significantly reduced disability at 90 days) of mechanical thrombectomy intervention [142]. This positive outcome was linked to the vessel’s abrupt recanalization and the ischemic zone’s quick reperfusion, following this model data [133,139].

Focal ischemia can also be induced by direct occlusion of the target vessel using a cranial window to clamp, ligate, or cauterize the vessel in situ, causing cortical and striatal lesions (Table 3B) [141]. Similar to the MCAO technique, focal ischemia can be induced either permanently or transiently [143]. After brief ischemia, this approach causes a sudden prompt mechanical perfusion [143]. The fact that this model involves assessing the target vessel through the skull, causing cortical spreading depressions and inflammatory reactions, limit its use in stroke research [143].

Since thromboembolism is a cause of stroke in humans, thromboembolic stroke models (Table 3C) may have considerable benefits: they may better reflect disease pathology [141], and offers the chance to evaluate thrombolysis therapy alone or in conjunction with protective drugs [144]. The reproducibility of thromboembolic stroke models and their usefulness for longitudinal research are negatively impacted by the variability in infarct location and volume; low survival rates; and [145] potential spontaneous clot formation after embolism disruption and spontaneous reperfusion [146,147].

Endothelin-1, a peptide with long-lasting action as a vasoconstrictor, is used in endothelin models (Table 3D) of ischemia to cause vascular occlusions [148]. Depending on the concentration of the peptide when applied, the degree and duration of the injury varies [141,149]. The slow lesion development, prolonged cerebral blood flow drop, and eventual progressive reperfusion profile match the progression of a clinical stroke [141,149]. Due to the difficulty in achieving constant diffusion, the topical delivery method of the peptide is a source of variation in this model [141,149].

The photothrombotic stroke model (Table 3E) causes localized permanent infarcts of the cortex by introducing a photosensitive dye (such as Rose Bengal) into the cardiovascular system. When the animal is exposed to a particular wavelength of light through the intact skull, the dye is activated, causing the formation of reactive oxygen species (ROS) that harm the endothelium. This causes platelets to activate and aggregate, forming a clot [141]. Small cortical lesions and low mortality associated with the model make it suited for longitudinal studies; however, the lack of penumbral tissue within the lesion reduces its translation into clinical practice, since the penumbra is the target tissue for protective strategies [150].

Animal models have been widely employed to investigate stroke etiology and the efficacy of potential therapeutic strategies [151,152]. However, because of the failure to translate promising data obtained in animal models into clinical practice, their relevance has been questioned [152]. One issue relates to long-term functional assessment [151,152]. Typically, histologic evaluation of lesion volume is performed as the *sole* indicator of the potential therapeutic efficacy of a drug in in vivo stroke models. This approach is an objective way to evaluate stroke outcomes, but it differs significantly from the endpoints employed in clinical settings [151,153,154,155,156,157]. The key measures in stroke patients is the functional outcomes using neurologic scores, such as the modified Rankin scale (mRS) and the NIHSS [153,154,155,156,157]. Multiple rodent studies have shown that lesion size does not always correlate with functional deficits and important behavioral outcomes, implying that lesion size alone is not sufficient to evaluate therapeutic success [153,154,155,156,157].

Thus, long-term studies are necessary to assess if the treatments’ effects on functional outcomes are significant, and ideally, a battery of tests should evaluate a range of functional parameters related to the damage, which has been suggested by the STAIR consortium for a long time [152,153,154,155,156,157,158]. As said above, the intraluminal suture of the MCA method has several recognized variables that can affect the resulting lesion and mortality, creating an ethical issue that weakens the reliability of the data [159]. Pinto et al. improved some relevant aspects of this method, including the optimization of the filament coating length, occlusion time, and postoperative care, resulting in mice with consistent and less variable lesion sizes, and with lower mortality rates [159]. This study also established a battery of functional tests to evaluate sensorimotor and cognitive functions similar to the assessment in human patients. These improvements are essential for the translation of preclinical studies into the clinical practice.

Another cause of this ‘translational roadblock’ is the low transparency in data presentation and failure in randomization/blinding of experiments, affecting the validity of experimental models [153]. Most bioscience journals offer little or no recommendations on what to include in animal research reports [160]. Animal numbers must be reported to assess the biological and statistical significance of the experimental data, to re-analyze the data, and for reproducibility [160,161,162]. To address this, ARRIVE (Animal Research: Reporting of In Vivo Experiments) guidelines were created with the CONSORT Statement as their cornerstone [160,161,162]. These include a checklist of items that describe the information to include in scientific publications reporting animal research, such as the number and specific characteristics of animals used (including species, strain, sex, and genetic background); details of housing and husbandry; and experimental, statistical, and analytical methods [160].

Preclinical stroke studies typically have low statistical power [163]. To conduct experimental research that complies with ethical standards and the 3Rs principles, it is necessary to use an adequate number of animals [141,164]. This prompted the scientific community to reevaluate experimental models and designs, to increase studies’ validity and translation into clinical practice [141,164].

## 3. Conclusions

It is expected that stroke, a complex and heterogeneous disease, continues to affect millions of people worldwide, as cases are going to increase dramatically. As such, it is imperative to find effective therapies, not only to increase survival rates but also to improve post-stroke outcomes. Nowadays, the treatments available rely only on reperfusion by mechanical thrombectomy or intravenous thrombolysis, which can only be applied in short time windows, leaving many patients without any treatment. However, there is a renewed sense of optimism in the field of stroke research, believing that new treatments will be discovered to benefit patients, as a result of better in vitro and in vivo models, rigorous experimental design, and closer interactions between clinical and preclinical research [165]. Additional evidence of the efforts being made to solve the translational roadblock can be seen in the recently published guidelines (Ischemia Models: Procedural Refinements Of in Vivo Experiments (IMPROVE)) created by a working committee led by the National Center for the Replacement, Refinement, and Reduction of Animals in Research (NC3Rs) [150].

Preclinical stroke research spent the last 50 years mostly focusing on developing acute neuroprotective medications; today, this research must comprehend the post-stroke consequences and develop strategies to reduce long-term disability [165]. A wide range of cytoprotective drugs have either undergone clinical trials or are presently being evaluated in pre-clinical research for efficacy in acute ischemic stroke, over the past five years [166]. An example of this is nerinetide, which showed promising results in reducing brain infarction and stroke mortality and improving patients’ functional outcomes in clinical trials [77]. Furthermore, a major step towards protection has been recently established, with the reconsideration that cytoprotectants might be administered together with EVT/IVT therapies [69,70]. Multiple previously discarded compounds are now being re-purposed under this new light and may provide promising preclinical results, and advance to clinical trials [94]. New imaging techniques provide innovative mechanisms to verify the efficacy of the tested cytoprotectants, and the development in the design of clinical trials brought improvements in the discovery of promising treatments [70].

The current challenge for stroke investigators is to pinpoint the mechanisms that targeted novel treatments while keeping up the quest for new approaches to reduce early damage [165]. Taking this into account, it is also important to choose an adequate model for each study, evaluating their strengths and weaknesses to overcome the existent translational roadblock. Thus, we summarized the developments made over the last few years regarding in vitro and in vivo models, taking into account the developments made in treatments and unraveling new or more suitable preclinical models in the ischemic stroke field to bring more effective treatments for such a devastating disease.

## Figures and Tables

**Figure 1 biomedicines-10-02561-f001:**
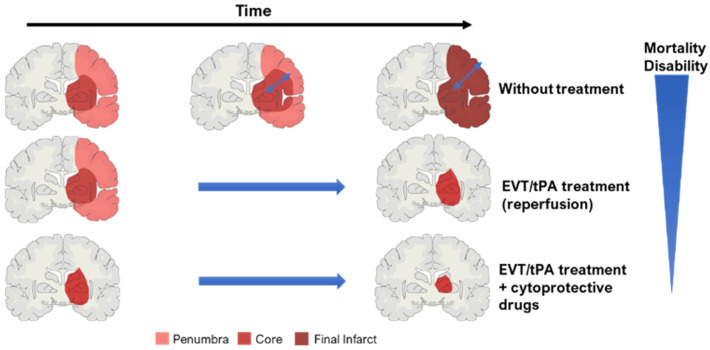
Concept of ischemic core and penumbra, and their progression through time (EVT—endovascular/mechanical thrombectomy; tPA—thrombolysis). The blue arrow within the brain in the ‘without treatment’ condition represents core expansion.

**Figure 2 biomedicines-10-02561-f002:**
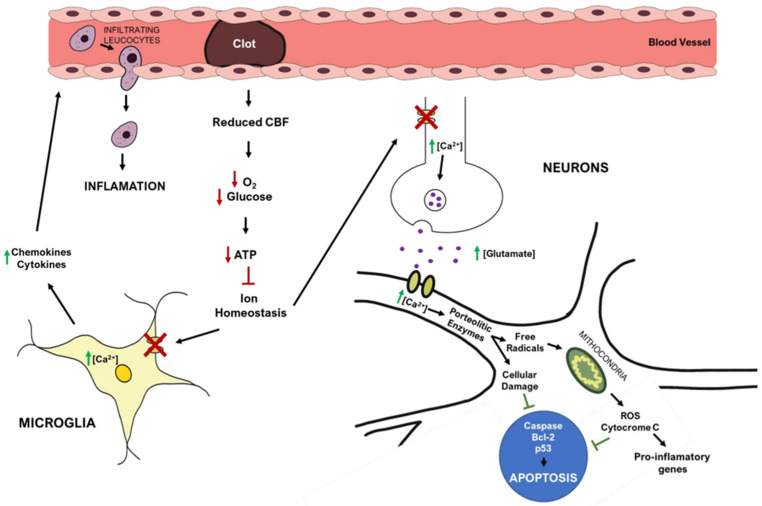
Simplified overview of stroke’s pathophysiology: molecular pathways occurring in the brain after ischemic stroke. Red arrows represent decreased events, while green arrows represent increased events.

**Figure 3 biomedicines-10-02561-f003:**
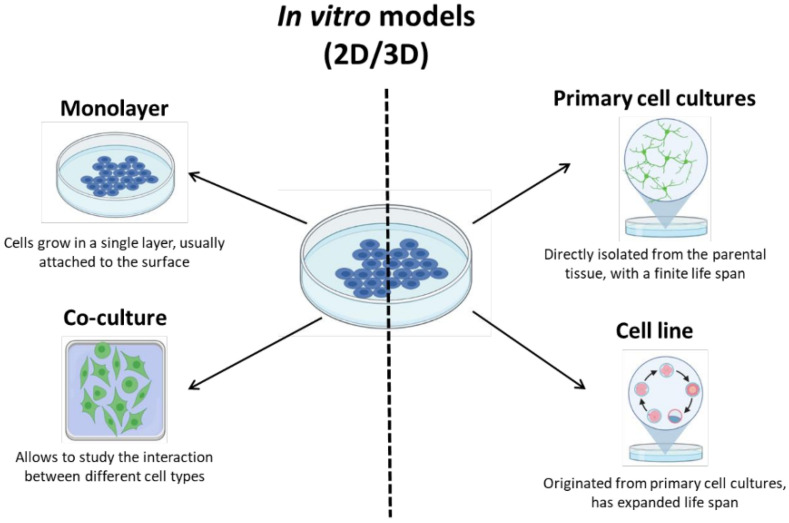
Culture of brain cells for 2D and 3D in vitro models.

**Table 1 biomedicines-10-02561-t001:** Overview of recent clinical trials for cytoprotective therapies—phase, status, and mechanism of action.

Clinical Trial	Therapeutic Drug	Mechanism of Action	Status of Clinical Trial
VENUS (Phase III)	Nimodipine	Blocks voltage-gated channels (calcium)	Terminated: not effective
MAVARIC (Phase I)	Verapamil (with magnesium sulfate)	Blocks of voltage-gated channels (calcium), after reperfusion	Results not published yet
ESCAPE-NA1 (Phase III)	Nerinetide (Na-1)	Inhibits neuronal excitotoxicity	Approved by American FDA alone; but it does not display protection together with IVT
ESCAPE-NEXT (Phase III)	Nerinetide (Na-1) together with EVT therapy	Inhibits neuronal excitotoxicity	Ongoing (completes in August 2023)
URICO-ICTUS (Phase III)	Uric Acid (UA)	Antioxidant agent (reduces inflammation)	Small study sample, but a confirmatory trial is planned
BAST (Phase III)	3-N-butylphtalide (NBP)	Free radical scavenger (reduces inflammation)	Ongoing (completed on 31 December 2022), good outcome so far
Evaluation of HUK in AIS (Phase IV)	Human Urinary Kallidinogenase (HUK)	Anti-inflammatory agents suppress TLR4/NF-kB pathway	Approved by China’s FDA, with post-approval surveillance ongoing
A study of allogeneic mesenchymal bone marrow cells in AIS (Phase I/II)	Allogeneic adult mesenchymal bone marrow cells	Cell-based therapy	Completed, presented beneficial behavioral outcomes for patients
J-REPAIR (Phase I/II)	JTR-161 (allogeneic stem cell product)	Cell-based therapy	Last updated on June 2022, results not published yet
Efficacy and safety of FTY720 for AIS (phase II)	Fingolimod (FTY720)	Immune modulator	Completed; reduced injury and improved clinical outcomes
Combination of the immune modulator Dimethyl Fumarate with alteplase in AIS (phase I/II)	Dimethyl Fumarate	Immune modulator	Ongoing (completes in December 2022)

**Table 2 biomedicines-10-02561-t002:** Stroke models in vitro: comparison of spheroids and organoids, in terms of culture and their advantages and disadvantages.

	Cellular Source	Culture Conditions	Advantages	Disadvantages
Spheroids	Differentiated cells, multicellular mixture, primary cells	Cultures with or without ECM and growth factors, self-assemble on heterogenous sphere format	○ Allow co-culture ○ Cost-effective ○ No expensive material ○ Create a gradient of pH, nutrients, and gas ○ Quick preparation, easy to maintain, suitable for HTS	○ Can be heterogeneous ○ Fragile structure ○ Complications with drug testing due to different gradients and density within the whole spheroid
Organoids	Only stem cells: embryonic, adult, or pluripotent	Require ECM and growth factors, self-assemble with different cell lineages, resembling the structure and function of an organ	○ Allow co-culture ○ Reproduction of cell–cell and cell–ECM interactions ○ Stable for long-term maintenance	○ Expensive ○ Require special equipment ○ Low reproducibility ○ Matrix dictates therapeutic response ○ Not suitable for HTS

**Table 3 biomedicines-10-02561-t003:** Stroke models in vivo: illustrations of stroke models with a schematic representation of the affected areas (core and penumbra), and advantages and disadvantages of each model.

In vivo Stroke Model	Advantages	Disadvantages
A. Intraluminal suture MCAO model 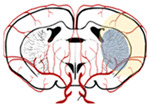	○Resembles human stroke in location○Mimics core and penumbra○Reperfusion highly controllable○No craniectomy	○Significant welfare effects include death, abnormal/reduced motility, weight loss, and difficulties eating and drinking○Filament use is an all-or-nothing strategy, not representative of the clinical presentation○Not suitable for thrombolysis studies
B. Electrocoagulation of MCA model 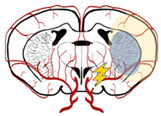	○Resembles human stroke in location○Uses the cranial window to directly occlude the MCA or vessel of interest○Visual confirmation of successful MCAO	○Permanent ischemia○Craniectomy
C. Embolic stroke model 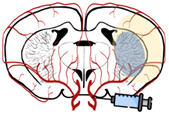	○Mimics more closely the pathogenesis of human stroke○Appropriate for studies of thrombolytic agents	○Spontaneous recanalization○High variability of lesion size
D. Endothelin-1 model 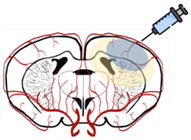	○Applied directly to the vessel of interest○Low invasiveness○Severity and duration of ischemia depend on ET-1 concentration	○Topical peptide administration serves as a source of variation○Low infarct reproducibility
E. Photothrombosis model 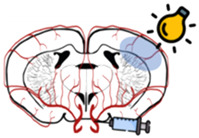	○Enables well-defined localization of ischemic region○Highly reproducible○Low invasiveness○Low mortality	○Lack of penumbra○Not suitable for investigating protective agents

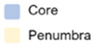
.

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
