# Peer review of "Ischemic Stroke, Lessons from the Past towards Effective Preclinical Models"

_biomedicines, 2022, doi:10.3390/biomedicines10102561_

Round 1
Reviewer 1 Report
This review is pretty comprehensive and the manuscript is well written. I don't see any major issues.
I think the manuscript was well written. It was a pretty comprehensive review, evidence based with good illustrations, tables and good selection of references. I didn't see any major issues, however, the tables might benefit from some fine tuning.
Author Response
We acknowledge the reviewer for the positive comments and for peer-reviewing our manuscript. Together with IJMS proofs team, we will also polish the tables as reviewer pointed.
Reviewer 2 Report
This is very good and interesting review paper regarding ischemic stroke. Since this is well-written and well-structured, I have only few things for its improvement as below.
1. Typo and grammatical errors are found in the text. Therefore, authors should check again thoroughly.
2. Future persfectives are not fully included in the Conclusion section. Please give some messages regarding future treatment way in the last context.
3. Authors need to list up new drugs which are under studying with new target in this field.
Author Response
We acknowledge the reviewer for the careful revision and taking time to peer-review our work.
We agree with the minor points indicated, and so we have revised the manuscript for typo and grammatical erros, as well as added future perspectives to the conclusion. Additionally, we looked at clinical trials platform, for newer drugs under investigation, and updated the list of drugs as suggested.
We think, with these revisions, the manuscript improved in quality, and is now ready for publication.